# Milk Consumption for the Prevention of Fragility Fractures

**DOI:** 10.3390/nu12092720

**Published:** 2020-09-05

**Authors:** Liisa Byberg, Eva Warensjö Lemming

**Affiliations:** Department of Surgical Sciences, Orthopaedics, Uppsala University, SE-751 85 Uppsala, Sweden; eva.warensjo.lemming@surgsci.uu.se

**Keywords:** milk, dairy, fragility fracture, hip fracture, observational studies, bias, nutritional epidemiology

## Abstract

Results indicating that a high milk intake is associated with both higher and lower risks of fragility fractures, or that indicate no association, can all be presented in the same meta-analysis, depending on how it is performed. In this narrative review, we discuss the available studies examining milk intake in relation to fragility fractures, highlight potential problems with meta-analyses of such studies, and discuss potential mechanisms and biases underlying the different results. We conclude that studies examining milk and dairy intakes in relation to fragility fracture risk need to study the different milk products separately. Meta-analyses should consider the doses in the individual studies. Additional studies in populations with a large range of intake of fermented milk are warranted.

## 1. Introduction

Milk and dairy products have long been promoted as part of a healthy diet, since they contain many essential nutrients (18 of 22), including calcium, phosphorous, and vitamin D [1]. Current dietary guidelines for milk and dairy consumption are based on the recommended daily intake of calcium needed to maintain a good calcium balance to prevent fragility fractures. Fragility fractures constitute a large and growing problem worldwide in both women and men. A hip fracture is the most devastating fragility fracture, with a profound impact on quality of life and mortality [2]. Fracture risk is influenced both by the genetic constitution and by environmental factors, where lifestyle factors (e.g., diet and physical activity) are becoming of greater importance with increasing age [3]. Modifiable lifestyle factors therefore provide interesting targets of primary prevention. 

The phrase “milk gives you strong bones” has been used in promotional campaigns both in Sweden and in the US. In contrast, ecological studies show that countries with the highest calcium and total milk product intakes have the highest fracture rates [4]. Different milk products are frequently considered as one entity in cohort studies. Sometimes because the information on different products is missing, sometimes to increase the span of intake when intakes from the separate milk products are small, and also, because they have been thought to have the same impact on health. Although milk and fermented milk products such as yoghurt and soured milk are similar in nutrient and protein contents, the fermentation process in the latter products starts by adding bacterial cultures of varying sorts to fresh milk. During the fermentation process, some of the carbohydrates in the milk are digested; the resulting product is often more viscous and may impact the postprandial response after ingestion. The bacterial culture itself may influence the gut microflora [5] and have anti-inflammatory and antioxidant effects [6,7,8], as discussed in more detail below. Thus, it is of importance to study different milk and dairy products separately, as will be highlighted in this review. 

In recent years, systematic reviews and meta-analyses of studies investigating the role of milk and dairy products in relation to fragility fracture risk have been performed and cover the available original research until 2019 [9,10]. In this narrative review, our aim is not to conduct another systematic review and meta-analysis but, rather, to discuss the available studies with regards to their relative merits, highlight potential problems with meta-analyses of milk and dairy intakes in relation to fracture risk, and discuss potential mechanisms underlying the different results. A comparison of cohort studies requires attention to population differences regarding setting, age, gender distribution, and length of follow-up. Additional attention to other important aspects is needed when cohort studies investigating dietary intakes in relation with fragility fractures are compared, and we will discuss these in the following. The studies included are mainly based on European and North American populations, and we only consider cow milk products.

## 2. Definition of Dairy and Milk Intake 

Dairy products are produced from the milk from milking animals. Cow milk is most widely produced and consumed worldwide and is the focus of the present paper. Liquid milk comprises the different types of milk commonly available to consumers (including pasteurised milk, skimmed milk, standardised milk, reconstituted milk, ultra-high-temperature milk, and fortified milk). Raw milk, as obtained from the cow without either addition to it or extraction from it [11], is consumed by humans in much lesser quantities due to legislative restrictions for sales because of the hazard of foodborne pathogens in unpasteurised milk [12,13,14] and is, therefore, difficult to examine in large population-based studies. 

Fermented milk is produced by the addition of bacterial cultures to fresh milk to start the fermentation process. Examples include yoghurt, kefir, dahi, Swedish filmjölk, the Norwegian kulturmelk, and the Finnish viili. The added sugar content varies highly, from the “natural” or unflavoured products to those with sugar, jam, fruit, or sweetener added to flavour the product, especially in yoghurt. Traditionally, the fluid remaining after the churning of butter has been saved and fermented into buttermilk. Such products, including piimä (Finland) and kærnemælk (Denmark), are now industrially produced, separate from butter production. 

Butter, cream, and fermented cream (sour cream and crème fraiche) are rich in fat, and the amounts consumed are generally small in comparison with the liquid and fermented milk products. Cheese is also produced from milk where the milk protein casein is coagulated and then separated from the whey. Since lactose is mainly found in whey, lactose and galactose contents in aged hard cheese are very low and can, therefore, be recommended also to galactosaemic patients [15]. The different types of cheeses and their characteristics depend on the type of processing, the type of milk and cultures used in the process, and the ageing period. Liquid milk, fermented milk, and cheese are the main dairy products consumed in Europe and the USA. In the following, we will refer to liquid nonfermented milk as “milk”.

The consumptions of milk, fermented milk, and cheese have changed over time, a pattern seen both in Northern Europe and in North America [16]. Figure 1 illustrates the dairy consumption in Sweden between 1950 and 2013, where milk consumption has decreased with a concomitant increasing consumption of fermented milk and cheese. Milk and dairy consumption is highest among children, intermediate among adolescents and teenagers, and lowest among adults, a pattern seen both in the US [17] and in Sweden [18].

Combining all milk product intake into a “total dairy” intake might be problematic because of secular trends, differences between populations in the proportions of different milk products, and different definitions of “total dairy”. The method used to assess the dairy intake and, thereby, also the kind of consumption included may differ and have an impact on the interpretation [19]. 

For the assessment of dairy consumption, most larger cohort studies use food frequency questionnaires (FFQs) that ask about the usual consumption during the past six months or year, inquiring about the number of glasses of milk or the number of servings of yoghurt and soured milk consumed per day or week. From the FFQ, it is usually difficult to obtain information on dairy products used as ingredients for cooking, and intakes might, therefore, not be directly comparable with information from a food record, where milk and dairy intakes also can include contributions from composite dishes (for example, pancakes). The total dairy exposure from such a study would then reflect both the milk consumed as a beverage and the dairy included in the composite dishes, which may be a problem, especially in populations where milk intake is low. 

It is likely that a long-term exposure before the fragility fracture occurs is needed, and the timing of when the dairy intake is measured, both in relation to biological age and period effects, will influence the intake range of the different milk products (Figure 1), and the influence of potential changes over time will depend on the length of follow-up. In a longitudinal study with repeated dietary assessments, it is possible to get a better estimate of the usual intake over several years that, consequently, also will reflect societal trends in dietary habits. Since the included exposures may have effects in opposing directions, the average effect will depend on how commonly the different dairy products are consumed in the studied population at that time. Constructing a “total dairy” variable is, therefore, a clear case of mixing apples and oranges, and we strongly suggest that different dairy products should be analysed as separate exposures. 

## 3. Fragility Fractures

Fragility fractures pose a large burden on both costs for healthcare and the individual in terms of quality of life and disability [20]. The most common types of fragility fractures include wrist, vertebral, and hip fractures. A hip fracture is the most devastating type of fracture, not only associated with large costs and risk of disability but, also, an increased post-fracture mortality [21]. The risk of most fractures increases linearly with age, although the risk of a hip fracture increases almost exponentially. Women are more likely to suffer a hip fracture than men [22], and the average age at the time of a hip fracture is approximately 80 years in men and 82 to 83 years in women [23,24]. Hip fractures occurring in persons below the age of 70 years are likely to be pathophysiologically different from hip fractures occurring later in life [3]. The lifetime risk of a hip fracture at age 50 is 16% among women and 6% among men in the USA and higher in Sweden and Norway, with corresponding lifetime risks of 25–29% and 9–13% [22]. The higher risk of a fragility fracture among women is likely due to a number of factors, including a longer life span, differences in bone geometry, a lower bone mass [25] and accelerated bone loss after menopause [26], a lower muscle mass, and a higher prevalence of impaired balance [27]. 

With increasing age, both skeletal muscle mass and bone mineral density (BMD) declines, influencing sarcopenia, the fall risk, and bone strength [28,29]. Even if a large proportion of the variation in BMD in middle-age seems genetically determined [30], the heritability of bone loss [31] and fractures [3] is modest at older ages, indicating that lifestyle factors, including dietary factors, become of greater importance with increasing age. Osteoporosis, the clinical definition of a low BMD, is a major risk factor for fragility fractures, although the majority of fragility fractures occur among those with a BMD in the normal or osteopenic range [32]. Fragility fractures are most often a consequence of a fall from standing height or less [33]. 

Some studies separate such low-energy fractures from higher-energy trauma fractures. It has, however, been recommended to retain also high-energy fractures among fragility fracture outcomes in observational studies [34,35]. A low BMD is associated with an increased risk of fractures at all sites, except facial fractures [36], and comparable increases in the risks of low-energy and high-energy trauma fractures are seen in association with decreasing bone density in elderly people (≥60 years) [34,35].

## 4. Potential Mechanisms

Milk and other dairy products constitute a major dietary source of calcium, vitamin D, and protein in the Western population. The importance of calcium and vitamin D for fragility fracture risk has been reviewed elsewhere [37,38,39]. Total, animal, and vegetable protein intakes were not associated with fracture risk in a recent meta-analysis [40], although a couple of studies have shown that intakes of dairy protein, but not vegetable protein, were associated with measures of bone strength [41,42]. Dietary protein supplements increase the skeletal muscle mass during resistance training, especially among those already trained and younger individuals [43]. As discussed above, both the bone strength and skeletal muscle mass are of importance for fracture risk. 

Whey and casein are the major proteins in milk. Although individual studies have suggested effects of whey protein on bone and skeletal muscle health, systematic reviews and meta-analyses conclude that whey protein supplementation does not improve such outcomes to an extent larger than that of other proteins [44]. Whey may have a beneficial effect on inflammation; however, the studies exploring the effect among older individuals are few [45]. Recently, genetically determined variants of casein, A1 and A2 β-casein, available in different proportions in different cow breeds [46], have been suggested to have differential effects on health, with A1 β-casein being associated with gastrointestinal problems and inflammatory markers [47,48]. However, the overall evidence for a major impact on health seems scarce [46,48]. The relative abundance of the β-casein variants is not largely different between the US and Europe, and dairy producers most commonly do not separate the milk by β-casein variant; differences in β-casein subtypes, therefore, are unlikely to explain different results from studies in the USA and Europe.

Interventional and observational studies indicate that milk intake is associated with higher concentrations of insulin growth factor-1 (IGF-1), a growth factor important for cell growth and metabolism [49,50]. In children, milk intake stimulates linear growth, especially among those with poor nutrition, and there seems to be an observational association between milk intake and height also in well-nourished populations [49]. Being tall is associated with a higher risk of hip fractures [51]. On the other hand, a high milk intake during childhood is associated with higher bone mineral density [52,53], which may provide a larger margin for osteoporosis when the age-related decline in bone mineral density begins. However, prospective studies on the milk intake in childhood in relation to fragility fractures are lacking. Although milk contains both IGF-1 and its binding proteins, it is unlikely that the absorption of these factors after the consumption of milk is the cause for the increased concentrations in the blood [54]. Instead, other mechanisms have been suggested, including an effect of branched amino acids present in milk proteins—for example, casein [54]. 

Several other components of milk have been linked with biological effects, including antioxidant effects [55,56]. Oxidative stress and low-grade inflammation may have an impact on the fragility fracture risk by influencing both the bone and skeletal muscle [57,58,59]. They may, therefore, be important mechanisms underlying the potentially different effects of nonfermented and fermented milk on the fragility fracture risk, since milk and fermented milk contain largely identical amounts of calcium, vitamin D, and proteins. 

Fermented milk is produced from fresh milk by the addition of bacterial cultures that starts the fermentation process. As a result, fermented milk has potential probiotic, antioxidative, and anti-inflammatory effects [6,7,8], reviewed in the following section. 

Bacterial strains in starter cultures for the fermentation of milk influence the microbiome, seen in animal and in vitro studies [5]. *Lactococcus Lactis* consumed in fermented milk is metabolically active and survives passage through the gastrointestinal tract [60] and may have an impact on the surrounding intestinal microbiome. A recent randomised cross-over study showed that a high-dairy diet compared to a low-dairy diet increased the abundance of lactate-producing bacteria and, at the same time, decreased the abundance of *Faecalibacterium prausnitzii*, a strain with several beneficial functions, including anti-inflammatory effects [61]. However, the proportions of fermented and nonfermented milk varied between participants. It is not clear to what extent nonfermented and fermented milk may have differential effects on the gut microbiome, largely because most available studies are small and compare different interventions. 

In the fermentation process, lactic acid bacteria produce lactic acid by the metabolism of lactose into glucose and galactose. Some of the monosaccharides are also consumed in the process. As a result, sour milk and yogurt have lower contents of lactose and galactose than milk [62,63]. The total galactose content (free galactose plus the galactose moiety of lactose) in natural yogurt is 90–95% of that in nonfermented milk, and the total galactose load in sour milk (filmjölk and Kefir) is 50–80% of that in nonfermented milk [62,63], depending on the storage time. The lactose and galactose contents in hard cheeses are close to null [64]. Galactose is mainly metabolised via the Leloir pathway, generating glucose. The enzyme galactose-1-phosphate uridylyltransferase (GALT) in the Leloir pathway is less active among women and is the most common defect in the congenital condition galactosaemia. Even with treatment, children with galactosaemia have high levels of oxidative stress and suffer from osteoporosis and an increased risk of chronic disease. One of the alternative pathways for galactose degradation, not going through the GALT enzyme, produces free radicals as a biproduct, and galactose reacts nonenzymatically with amino acids, resulting in advanced glycation end-products, as overviewed in [65]. The induction of oxidative stress and chronic inflammation by giving D-galactose is an established animal model of ageing [66,67,68]; a dose corresponding to one-to-two glasses of milk in humans accelerates senescence in rodents [68]. 

Since oxidative stress occurs when there is an imbalance between free radicals and scavenger mechanisms, an increased free radical production might be counteracted when fermented milk products are consumed by the effects of lactic acid bacteria on the microbiome, degradation of galactose, and other potential antioxidant effects. Fermented milk products with probiotics have been shown to reduce the levels of inflammation markers [69]; other studies have been too small to show conclusive effects [7,70], and the results may also be dependent on the probiotic strain [71]. In our studies, a high intake of milk was associated with higher concentrations of inflammation and oxidative stress markers, whereas high intakes of sour milk and yoghurt were associated with lower concentrations of such markers [72].

It is likely that multiple mechanisms are involved in how milk and fermented milk influence the risk of fragility fractures. The effects may both strengthen and have deleterious effects on the bone and skeletal muscle directly or indirectly via, for instance, decreased or increased inflammation and oxidative stress. The net effect on the fragility fracture risk will be the average of these mechanisms and is likely to depend on both the type of milk (nonfermented or fermented) and amount ingested.

## 5. Interventional and Observational Studies

Randomised controlled trials (RCTs) of dietary interventions on fragility fracture risk are scarce. The complex interactions of dietary components in the food matrix [73], and that food intake is part of everyone’s daily life, makes blinding, long-term interventions or restrictions, and compliance to the interventions more difficult in dietary interventions compared to pharmaceutical interventions [74]. RCTs of different milk products on intermediate outcomes can provide important insights into short-term biological effects that may influence fragility fracture risk. The effects on fragility fractures are, however, likely to be the result of long-term exposures. The long-term effects of milk and dairy products on hard endpoints such as fragility fractures are, therefore, unlikely to ever be studied in an RCT. The field of nutritional epidemiology includes methods to deal with these issues [74] and relies largely on observational studies. 

It seems especially important to assess the intake of milk and dairy products before the fragility fracture occurs, since the fracture itself may influence how this is reported [75]. Estimates from case-control studies where the information has been collected post-fracture may, therefore, be biased. Due to these limitations, this review will focus on cohort studies.

## 6. Meta-Analyses of Cohort Studies Examining Milk Intakes and Hip Fracture Risk

Most meta-analyses have focused on either the total dairy intake or milk intake in relation to the risk of hip fracture, the major fragility fracture. Only a few cohort studies thus far have reported separately on other dairy products. As indicated above, the total dairy intake may be of less importance, since the overall effect on hip fracture risk may be dependent on how commonly different dairy products are consumed in different populations. The discussion and examples below will therefore focus on meta-analyses of studies examining the association of milk consumption with hip fracture risk.

For many research questions, meta-analyses of randomised controlled trials or, if not feasible, of cohort studies provide the best evidence. By combining estimates from several studies of varying sizes, a higher statistical power is obtained, and the weighted average of the available data is considered as a more robust point estimate than the estimates from the individual studies [76]. 

Several meta-analyses of milk intake on hip fracture risk have been presented, and most conclude that there is no overall association. In meta-analyses of drug interventions with strictly standardised interventions set to a fixed dose, the variability lies mostly in population differences, such as the age and sex distribution of the population studied, the intervention time and follow-up, and the outcome assessment. Meta-analyses of nutritional observational studies are challenged with additional aspects and may, as a result, have an increased variability and reduced possibility to detect real effects [77]. In the following section, we discuss some of the aspects that potentially disqualify the results of meta-analyses of milk intake in relation to hip fracture risk. The heterogeneity of studies causes problems when performing a meta-analysis, as also discussed in the systematic review on milk intake and mortality by Larsson et al. [78].

The main result from a meta-analysis is commonly presented as the relative risk (RR) of hip fracture in the highest category vs. the lowest category of milk intakes or as the RR of a hip fracture per 200 g milk per day. In recent meta-analyses of cohort studies, these RR estimates were 0.91 (95% CI: 0.74–1.12) for the highest vs. lowest milk intakes [9], 1.00 (0.94–1.07) [9] and 0.93 (0.75–1.15) [10] per 200 g milk/day, and 1.09 (1.07–1.11) per 200 g milk/day using meta regression [10], indicating a lower risk of hip fracture, no association, or a higher risk of hip fracture with higher milk consumptions. What, then, do these estimates represent? The estimate per 200 g of milk per day indicates a linear association of the milk intake and RR of hip fractures over the full range of milk intakes in the meta-analysis estimated from each included study. The meta-analysed estimate comparing the highest vs. the lowest category of milk intakes will equate the estimates from studies with different exposure ranges. 

In Table 1, the cohort studies included in the meta-analysis by Bian et al. [9] are listed, together with a recent study from the USA [79] that was additionally included in the meta-analysis by Malmir et al. [10]. This meta-analysis [10] also included a smaller study with 43 hip fractures that does not clearly specify the amount of milk in the tertiles used in the analysis [80] and an updated analysis [81] of the women in reference [72], resulting in duplicate observations being included. One recent study from Norway [82] that could not separate between nonfermented milk and fermented milk and one from Australia [83] using major osteoporotic fractures as the outcome were not considered. 

The highest and lowest categories of milk intakes are listed in the second column of the table. For example, in the studies by Fujiwara [84], Owusu [85], and Sahni [86], the highest category of milk intakes ranged between two-and-a-half and at least seven glasses/week, corresponding to 72–200 g or up to one glass milk/day. In contrast, the lowest category of milk intake was 200 g (one glass) milk/day in several of the other studies [72,87,88], and the highest intake categories ranged between 200 and 1000 g (three to five glasses) milk/day. The effects of milk intake on the hip fracture risk in the lower range of milk intake are, thus, equated with the effects in the upper range of milk intake, with the reference categories being completely different. This also means that, for studies in populations with a low milk intake—for instance, where 200 g milk/day is a high or maximum intake—the estimate comparing one glass per day vs. no intake will be extrapolated over the full range of the milk intake distribution, even if no one in that population consumes that much milk. Thus, combining estimates based on the highest vs. lowest intake categories or per 200 g milk/day should be avoided unless the milk the intake distribution is similar in all included studies. 

**Table 1 nutrients-12-02720-t001:** Cohort studies of milk consumption in relation to hip fractures: milk intake categories, hip fracture ascertainment method, number of hip fractures, and socioeconomic status (SES) adjustment.

Study: First Author (year)	Highest and Lowest Milk Intake Category	Hip Fracture Ascertainment	Number of Hip Fractures	SES Adjustment
Cumming (1997) [89]	3 glasses/day vs. rarely/never	Self-report ^a^	306	No
Fujiwara (1997) [84]	≥5 vs. 1 glass/week	Self-report + medical records ^a^	55	No
Meyer (1997) [88]	≥5 vs. 1 glass/day	Self-report + medical records	213	Yes
Owusu (1997) [85]	2.5 vs. ≤1 glass/week	Self-report ^a^	56	No
Kanis (2005) (meta-analysis) [90]	Highest vs. lowest	Self-report/registers	413	Not reported
Feart (2013) [91]	Highest vs. lowest	Self-report	57	Yes
Feskanich (2014) [87]	≥4 vs. 1 glass/day	Self-report ^a^	1716	No
Michaëlsson (2014) [72]	≥3 vs. <1 glass/day	Registers	5425	Yes
Sahni (2014) [86]	≥7 vs. 1 glass/week	Self-report + medical records	97	No
Feskanich ^b^ (2018) [79]	≥480 mL/day vs. <240 mL/week	Self-report + death records ^a^	2832	No

^a^ Report exclusion of high-energy trauma fractures. ^b^ Study not included in the meta-analysis by Bian et al. [9].

To circumvent the problem of comparing estimates based on different parts of the exposure range of milk intake, dose-response analyses or meta regressions can be performed. In contrast to the estimates of 0.91 (95% CI: 0.74–1.12) for the highest vs. lowest milk intakes and 1.00 (0.94–1.07) per 200 g of milk per day presented as the main results by Bian et al. [9], the dose response analysis in the same study, based on 11 effect sizes from eight cohort studies, indicated a relative risk of 1.09 (95% CI 1.07–1.11) per 200 g of milk/day, similar to the meta-regression results reported by Malmir et al. [10]. Compared to no milk intake, the meta-regression indicated a nonlinear association with hip fracture risk, where intakes up to 200 g of milk daily were associated with lower risk, and intakes above that were associated with higher risk [10]. Dose-response analyses and meta-regressions compare effect sizes at the same levels of milk intakes and are, therefore, better representations of the overall association; a study where the population only consumes low amounts of milk will only contribute to the estimate in that range of the milk intake distribution.

Meta-analyses using a dose-response analysis or meta-regression to estimate the association of milk intake in relation to hip fracture risk thus indicate that a higher intake of milk is associated with a higher risk of hip fracture [9,10]. In contrast, a higher intake of fermented milk may be associated with a lower risk of hip fracture, although the number of studies including this information is limited, and intakes are, in general, low [9,79,81]. Milk and fermented milk intakes also display associations in opposite directions in relation to mortality in several Swedish populations, where the range of intakes of both milk and fermented milk are wide [65,92,93].

## 7. Hip Fracture Ascertainment Method (Outcome Assessment)

Healthcare systems around the world have different ways of recording diseases in populations. Some countries, including Denmark, Sweden, and Finland, have national patient registers to which it is mandatory to report all hospital admissions, and there is the additional possibility to link this information to other data using personal identifiers unique to each individual. Hip fractures are almost exclusively treated in hospital and, therefore, are reported to such registers. The validity of such registers for capturing diseases will vary depending on the outcome and how the disease is treated [94]. In a cohort study with the possibility to link individual baseline data completely to in-patient registry data, there will be no or minor losses to follow-up when considering hip fractures as the outcome.

Other countries, such as Norway [95], have specific hip fracture registers or studies aimed at capturing as many hip fractures as possible, whereas many countries, including Italy and the USA, lack national registries with the possibility to link individual information. When linkage is not possible, a follow-up of participants in a cohort study needs to be based on other methods, including the tracking of the participants’ health records or follow-up by phone or mail where the participant self-reports any hip fractures and other disease outcomes. Both such methods rely on that these persons can be tracked even if they change care provider or move to a new address.

Self-reported hip fractures are largely accurate when the information reported has been compared with hospital records of the event [96,97]. However, the issue with self-reported hip fractures as an outcome in cohort studies lies in those fractures not reported, as elaborated on below. Mortality after a hip fracture is high; in 1986–2004, the one-year mortality after a hip fracture event was 32–41% among men and 22–24% among women in the US [98,99]. Even in studies with a high participation rate (86%), it has been shown that self-reporting detects less than 60% of hip fractures when compared with discharge register data [100]. Furthermore, despite the self-reporting of hip fractures every fourth months in the Study of Osteoporotic Fractures, one out of five hip fractures were not reported [101]. When hip fractures are retrieved by self-report every two years, as in the Nurses’ Health Study (NHS) and Health Professionals Follow-up Study (HPFS) [79], the problem with the underreporting of hip fractures is likely to be larger. Information on hip fracture events are sometimes enriched with information from death certificates [79]. However, the proportion of hip fractures reported as a contributing cause on death certificates is most likely underestimated [102,103]. Not only the high post-hip fracture mortality but, also, the fact that many patients are discharged to nursing facilities after suffering their hip fracture [98] contribute to the low detection rate by self-reporting [100,101].

As an example, the hip fracture rate among women in the NHS was 1.3 per 1000 person-years at risk [79], more than three times lower compared to that of women in the Swedish Mammography Cohort (SMC) (4.2 per 1000 person-years at risk) [81], although the NHS and the SMC are comparable in terms of age, body mass index, and follow-up time. This difference is larger than expected, where the hip fracture rate at age 80 among women is 1.3 times lower, and the estimated lifetime risk of hip fracture at age 50 is 1.8 times lower among women in the USA compared to Sweden [22].

The method used for the ascertainment of hip fractures in the cohorts included in the meta-analysis by Bian et al. [9] are listed in the table. In the meta-analysis by Malmir et al. [10], the hip fracture ascertainment method contributed to heterogeneity (the hip fracture ascertainment method stated was, however, not correct for at least one of the studies included [81]). A stratified analysis indicated that, for self-reported hip fractures, the association of the overall per 200 g/day milk intake (not using meta-regression) with hip fractures was inverse, and for other methods of hip fracture ascertainment, the RRs were close to 1 [10].

## 8. Cohort and Population-Specific Characteristics and Confounders

As always when comparing results from different studies, it is important to consider the characteristics of the population studied. For research evaluating the potential effects of milk and dairy intakes on hip fracture risk, the sex and age distribution and time of follow-up are two major aspects. The question is also when the causal effect of the exposure influences the risk of hip fracture, and, therefore, the timing of exposure assessment and length of follow-up are important to consider, as discussed above.

The adjustment for confounders to obtain as good an exchangeability as possible between exposure groups is essential in all observational studies. The identification and selection of confounding factors should be done appropriately so that factors on the causal pathway are not included. It has been suggested that adjustments for dairy sources of, for example, calcium, vitamin D, and proteins provide more valid estimates for the effect of milk consumption on the fragility fracture risk than if they are not adjusted for [104]. However, these factors should be considered to be mediators of a potential effect, and adjustment for such is not recommended if the total effect of milk intake on the fracture risk is explored [105].

The bias can be more problematic if important confounding factors—for example, socioeconomic status—are not considered. The socioeconomic status is an overall term that may include the attained educational level, marital or cohabiting status, individual or household income, and class. Each of these aspects of the socioeconomic status can have a different influence on the dietary intake, other lifestyle factors, and the risk of disease. In Scandinavia and Europe, the total milk and dairy intake have traditionally not been associated with any aspects of socioeconomic status [106]. In contrast, milk intake, as a part of adherence to dietary recommendations, is strongly associated with a higher income and higher social class in the USA [107].

Of the included studies in the review and meta-analysis by Bian et al. [9], only a few adjusted their estimates for socioeconomic status (Table 1). Of note is that the NHS and HPFS studies from the USA do not take socioeconomic status into account [79,87]. A common argument is that the examined populations are homogenous with regards to educational level at the baseline (they are all nurses or health professionals). However, there will naturally be a variation in the total household income and socioeconomic status also within these populations, especially during a long follow-up. As shown in the NHS, the household socioeconomic status and individual changes in socioeconomic status during the long follow-up time are likely to influence dietary habits and the risk of different health outcomes, despite a homogenous educational level at the baseline [108]. A dietary supplement use, which is also linked with milk intake, is higher not only among those individuals with a higher educational level but is also correlated with the educational level of the head of the household and with the total family income [109]. As an example from the NHS, calcium supplement use was higher among those with very low consumptions of milk, but among high milk consumers, vitamin D supplement use, the intake of fish, and physical activity were higher, and the proportion of smokers was lower, suggesting a trend of a healthier lifestyle with a higher milk intake [79].

Death after hip fracture is also more likely among men and women with high comorbidity and low socioeconomic status [110,111]. The failure to take the socioeconomic status into account may therefore be problematic, both from a residual confounding perspective and from a differential loss to follow-up perspective. Analytical adjustments for socioeconomic changes among responders cannot reduce the problem of differential reporting of hip fracture events in nonresponders, as discussed in the section below.

It has been suggested that the heritability of hip fractures or underlying diseases, such as gastrointestinal problems, might contribute to both a higher milk intake and a higher risk of hip fracture. Men with at least one parent experiencing a hip fracture did not have a different milk intake compared to men who did not have a parent with a hip fracture [72]. Similarly, women who experienced a fracture or other comorbidities between two FFQ assessments did not increase their milk consumption compared to women who did not experience a fracture or comorbidity in this period [72,112].

In observational studies, having a treatment with proton pump inhibitors (PPI) to reduce stomach acid production is associated with an increased risk of fragility fractures [113,114]. A simultaneous high milk intake to relieve gastrointestinal problems has been put forward as an explanation of observational associations where a high milk intake is associated with a higher fracture risk. However, since patients prescribed PPI therapy tend to be more frail and have more risk factors for fractures than those not given these drugs, it is unlikely that PPI therapy is an independent risk factor for fractures [115]. Furthermore, bone mineral density was higher in a group that received PPI along with risedronate (a bone resorption inhibitor that increases the bone mineral density among those with osteoporosis) compared with those on risedronate alone in an RCT [116]. In our data, there was no indication that high consumers of milk had been prescribed PPIs more frequently than low consumers of milk [117].

## 9. Epidemiological Considerations—The Combined Effect of the Different Biases

In the previous sections, we have described differences between cohort studies that contribute to the varying results for the association between milk or dairy intakes and hip fractures. It seems likely that different dairy products display associations in different directions with hip fracture risk, and therefore, the association of the total dairy intake with hip fractures will strongly depend on the proportions of the different dairy products in a cohort, and exposure effects may cancel them out. In the following section, we will therefore again focus on the potential effects of milk intake on the hip fracture risk.

A loss to follow-up or failure to capture all hip fracture occurrences in a population, as in studies where hip fractures are self-reported, will lead to an underestimation of the number of hip fractures. If this loss to follow-up is equal across the entire distribution of milk intake, this will result in more uncertainty in the estimates, but the estimate in itself will not be biased. However, if we believe that the milk intake actually affects the hip fracture risk, the loss to follow-up will be differential; more hip fracture cases will be lost among those who drink more milk, resulting in bias [118]. Whether a fracture is reported will naturally depend on whether the person survives after the hip fracture, potentially adding to the bias. This situation is illustrated in the causal diagram in Figure 2a.

Causal diagrams, or directed acyclic graphs, have emerged as a useful tool for epidemiologists [119]. They illustrate the underlying causal assumptions where an arrow indicates a causal effect of one factor on another. They can assist in the selection of confounders to adjust for in analyses and in the interpretation of results by highlighting and providing an understanding of the effects of biases that may occur in observational studies.

To further complicate the causal diagram in Figure 2a, previous studies have shown that a high milk intake is associated with mortality, thus introducing an arrow from milk intake to mortality shown in Figure 2b. Further, the socioeconomic status affects the milk intake in some populations and influences the risk of both hip fracture and mortality (Figure 2c).

Studies relying on self-reported hip fractures will only be able to study those fractures reported (or captured on death certificates). In the causal diagram theory, this implies conditioning on the ascertainment of hip fractures. In the causal diagrams in panels Figure 2b,c, mortality is a collider with arrows pointing into the node (mortality) from hip fracture, milk intake, and socioeconomic status. Conditioning on a collider (here, mortality), or a descendant of a collider (here, self-reported hip fractures) will, as stated in rule 3 of the *d*-separation criterion in causal diagrams [120,121], introduce selection bias in at least one of the strata.

The resulting collider stratification bias, known as Berkson’s bias since the 1950s [118], is special in that it will draw the association between exposure (milk intake) and outcome (hip fracture) downwards. Even in the case of no association between exposure and outcome (RR = 1), the collider stratification bias may induce an inverse association (RR < 1). If the milk intake increases the risk of hip fracture (RR > 1), the estimate will be drawn towards the null, or, if the bias is stronger than the effect, RR may even be estimated as <1. The heterogeneity of the results illustrated when stratified by hip fracture assessment type indicated a lower RR for studies with self-reported hip fractures [10]. The problem with this type of bias has recently been discussed and exemplified by the birthweight paradox where smoking among pregnant women is associated with a lower infant mortality in their outcome when the analysis is restricted to low-birthweight infants [122].

Figure 2c also illustrates that socioeconomic status is a confounder for the association between milk intake and hip fracture, and the failure to take that into account will result in residual confounding, as discussed above. This could be resolved by additional adjustments. However, the problem with differential loss to follow-up and the resulting collider stratification bias cannot be dealt with by adjustment, because it is an inherent problem of the data. The authors of original studies, reviews, and meta-analyses should acknowledge this inherent problem with self-reported hip fractures and discuss the potential influence the bias might have on their estimates. According to the STROBE (Strengthening the reporting of observational studies in epidemiology) guidelines [123], acknowledging a bias is not enough; also, the potential effect on the estimates should be discussed. The ascertainment of hip fractures that does not depend on self-reporting will most likely not have the same risk of collider stratification bias. The loss to follow-up in register data is either negligible (as in Sweden, Denmark, and Finland) or will not be linked with the milk intake or socioeconomic status (as, for instance, hospital-based hip fracture registers in Norway).

**Figure 2 nutrients-12-02720-f002:**
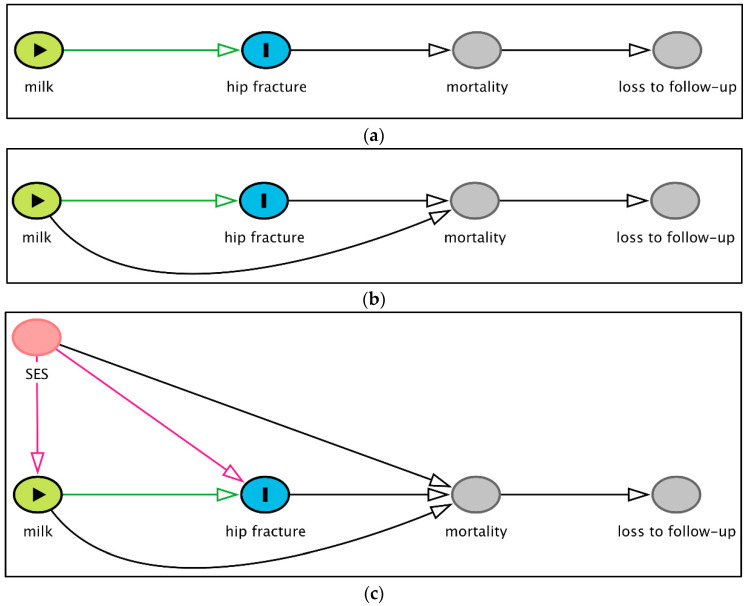
Causal diagrams indicating the assumed causal effects and illustrating the potential collider bias (Berkson’s bias) if hip fractures are ascertained by self-reporting. Since mortality is high following a hip fracture, not all hip fractures may be captured by self-reporting. Conditioning on a collider (here: mortality or, rather, survivors) or a descendant of a collider (here: loss to follow-up or hip fracture ascertainment) may induce a special form of collider bias, Berkson’s bias, known to bias associations between the exposure (milk intake) and outcome (hip fracture), so that a positive association is moved towards a null association, and if the bias is stronger than the association or if no association exists, it may even induce an inverse association. (**a**) Milk intake affects the risk of hip fracture, which, in turn, influences the risk of mortality and, thereby, also the probability of the hip fracture being reported by self-reporting (loss to follow-up). (**b**) Assuming the milk intake affects the risk of both hip fracture and mortality. (**c**) Assuming the socioeconomic status (SES) affects the milk intake, hip fracture, and mortality. The causal diagrams were constructed using DAGitty.net [124].

## 10. Mendelian Randomisation Studies

To overcome potential problems with residual confounding and reverse causation, genetic variants for lactase persistence have been used in Mendelian randomisation (MR) studies to explore the causal effect of milk intake on several outcomes, including hip fractures. Mendelian randomisation is the special case of an instrumental variable analysis where one or more common single nucleotide polymorphisms (SNP) are used as the instrument [125].

Lactase persistence is the ability to enzymatically split the milk disaccharide lactose into glucose and galactose and is genetically determined. Lactase (or lactase-phlorizin hydrolase) is encoded by a single gene, *LCT*. The prevalence of lactase persistence varies across populations but is high in Northern Europe; 89–96% of the population in Scandinavia and the British Isles are estimated to be lactase-persistent [126]. The inability to degrade lactose results in gastrointestinal problems, and those with lactose intolerance (or lactose nonpersistence) often limit their intake of milk products. However, the composition of the diet and the gut microbiome may influence lactose digestion [127] independent of the lactase genotype. Subjects with a lactose intolerance who experience gastrointestinal problems after consuming milk may better tolerate fermented milk products and compensate for the lower milk intake by consuming more hard cheese that contains almost no lactose [15].

The T-allele of a common C/T SNP (rs4988235) located in the *MCM6* gene 13910 base pairs upstream of the *LCT* gene determines the lactase persistence by regulating lactase expression [128,129]. It is associated with a higher milk intake in European-descent individuals and has, therefore, been used as an instrument in MR studies [130,131,132].

Only one published study has, to date, performed an MR study of the *MCM6* C/T SNP in relation to incident hip fractures as the outcome. The study included 97,811 men and women from three Danish cohorts [132]. The average age at the baseline in the included cohorts was 56–58 years, and 2121 incident hip fractures occurred during follow-up. The average age at hip fracture was not reported. Having the T-allele was associated with a higher milk intake, and the hazard ratio of hip fracture per T-allele was 1.01 (95% CI 0.94–1.09), indicating no causal effect of milk intake on the hip fracture risk. A meta-analysis of Danish and Northern European studies using any fracture as the outcome was inconclusive with odds ratios below 1 but with wide confidence intervals and large heterogeneity [132].

It may be argued that MR studies would be superior to cohort studies and prove that observed associations are biased and that there is no causal effect of milk intake on the hip fracture risk in this Danish population. However, MR studies also rely on strong assumptions that, similar to the assumption of no residual confounding in cohort studies, largely are untestable. The underlying assumptions of an MR study are illustrated in Figure 3. Note that, in a causal diagram, the absence of arrows between factors indicate that they are independent of each other.

For an instrument in an MR study to be valid [133], the instrument firstly needs to be associated with the exposure under study—in this case the milk intake, indicated by the arrow from the lactase persistence genotype to milk intake (Figure 3). Although having the T-allele was associated with a higher milk intake, those without the T-allele also consumed milk; the median intake among those with the lactase persistence (TT or TC) genotype was five glasses/week (interquartile range (IQR): 0–10 glasses/week) compared to three glasses/week (IQR 0–7 glasses/week) among those with the lactase nonpersistence (CC) genotype [132]. In a study from Finland, the prevalence of the CC genotype, indicating genetically determined lactose intolerance, was similar among those with self-reported lactose intolerance (15%) and among those with self-reported lactose tolerance (18%) [134].

Other studies have also indicated that the *MCM6* C/T SNP (rs4988235) is a weak instrument for milk intake [135] and might, therefore, induce bias in the presence of confounders [136]. The magnitude of the weak instrument bias depends on the strength of the association between the genetic instrument and the exposure, and the bias is in the same direction as the observational, potentially confounded association [136]. Furthermore, the dose response relation with milk intake is not readily observed using an MR study and nor is the consideration of type of dairy product consumed. Again, as discussed in this review, it is important to consider the different dairy products separately due to the potentially different effects on the fragility fracture.

Secondly, the instrument must not affect the outcome—in this case, the hip fracture—through any other mechanism than that through the exposure (milk intake), indicated by the fact that the only the path from the lactose persistence genotype to hip fracture in Figure 3 goes via the milk intake. Failure to do so induces what is called horizontal pleiotropy [137]. Thirdly, the instrument may not be affected by any other factors or confounders, indicated by the lack of an arrow from the confounders to the genotype (Figure 3). However, it has been shown that the lactase persistence genotype is not only related to the milk intake but, also, to fruit and vegetable and fish consumption (7), fermented dairy consumption (4), and to socioeconomic status in a way that cannot completely be explained by population stratification (8). These are all factors that are related to the fracture risk in observational studies, and such pleiotropic effects [138,139] violate the main assumptions of the MR study, and estimates cannot be interpreted causally, as illustrated in Figure 4.

In summary, it is unlikely that MR studies using the lactase persistence genotype as an instrumental variable will provide substantial evidence for the causal effect of milk on the fragility fracture risk.

## 11. Author Autonomy from Dairy Industry

It is important that research on the potential effects of different milk products also is performed by researchers without links to the dairy industry. The corporate funding of food and nutrition has repeatedly in most cases shown results that point in a direction that is beneficial for the industry sponsor [140]. The potential bias by the corporate funding of trials or even speaker fees or travel grants should not be neglected [141,142,143]. Of 22 review articles (in English, found in PubMed by the search term “review AND (milk OR dairy) AND (osteoporosis OR fracture)” and limited to those papers with milk, dairy, or other dairy products in the titles published in 2016–2020) published in the past years, 13 papers [144,145,146,147,148,149,150,151,152,153,154,155,156] were written by at least one author with links to the dairy industry or dairy associations, whereas the authors of nine papers [4,9,10,157,158,159,160,161,162] did not report any such conflicts of interest. Although not explored, the proportion of opinion papers, comments, and letters to the editors with conflicts of interest is likely to be higher.

## 12. Summary and Recommendations for Future Research

In summary, studies examining milk and dairy intakes in relation to the fragility fracture risk need to study the different milk products separately and need to have a good comparator [163]. More studies in populations with a large range of intakes of fermented milk are warranted. When comparing and combining the results from different observational studies in meta-analyses, the dose and exposure ranges should be considered and the fracture ascertainment method reported. This is in addition to the common evaluations of the age and sex distributions, confounders included, baseline risk of fragility fracture, length of follow-up, and proportion lost to follow-up. For a better understanding of the mechanisms and intermediate effects, short-term randomised trials are of importance, but it is unlikely that a long-term intervention study with restrictions to the type of milk and spanning over several years with fragility fractures as an outcome will ever be performed.

## Figures and Tables

**Figure 1 nutrients-12-02720-f001:**
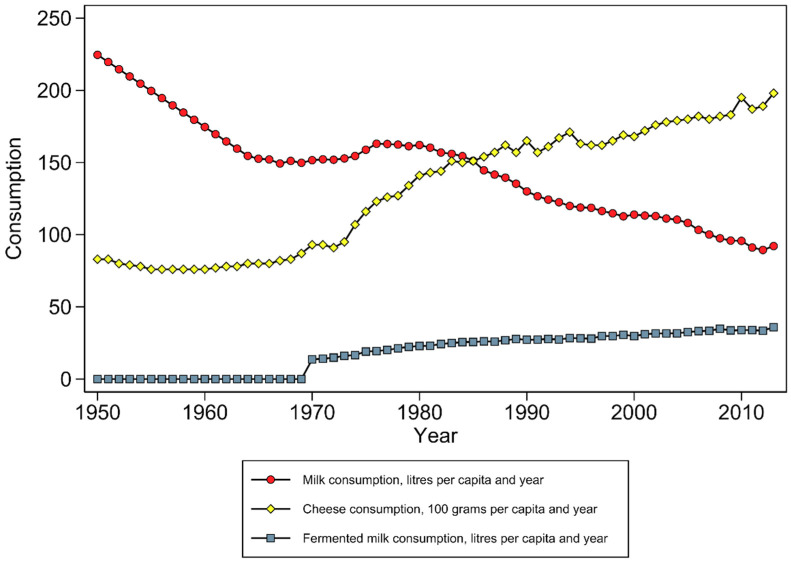
Consumption of milk, cheese, and fermented milk products in Sweden 1950–2013. Numbers from Jordbruksverket (Swedish Board of Agriculture) and LRF Mjölk (The Federation of Swedish Farmers, Milk Branch; previously, Svensk Mjölk).

**Figure 3 nutrients-12-02720-f003:**
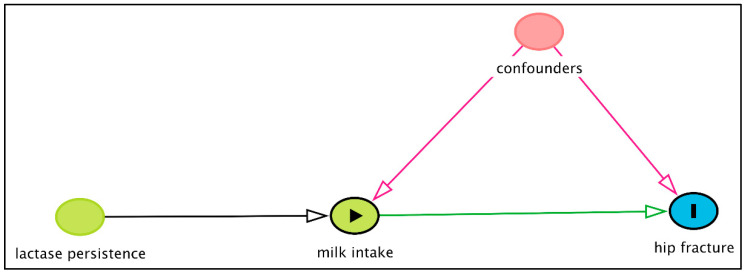
Causal diagram illustrating the underlying assumptions for a Mendelian randomisation study of the causal effect of milk intake on hip fractures using the *MCM6* C/T single nucleotide polymorphism (SNP) (rs4988235), here denoted as “lactase persistence”, as an instrumental variable for the milk intake. The causal diagram was created using DAGitty.net [124].

**Figure 4 nutrients-12-02720-f004:**
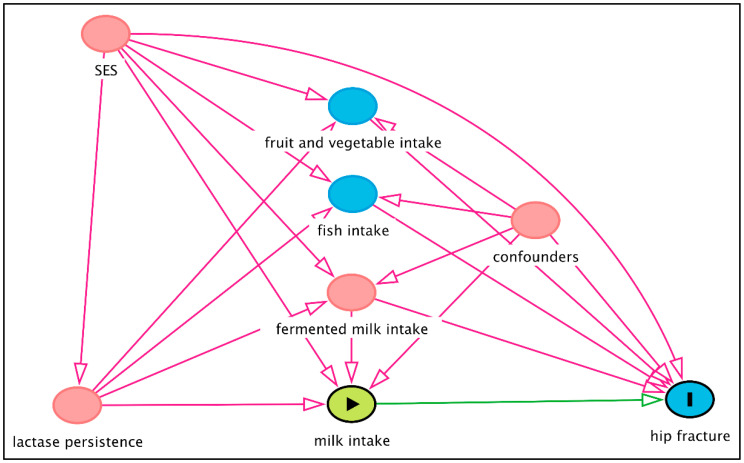
Causal diagram illustrating violations of the underlying assumptions for a Mendelian randomisation study of the causal effect of the milk intake on hip fractures using the *MCM6* C/T SNP (rs4988235), here denoted “lactase persistence”, as an instrumental variable for the milk intake: probable horizontal pleiotropic effects of lactase persistence through other biological pathways and potential confounding by socioeconomic status (SES). The causal diagram was created using DAGitty.net [124].

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
