# Peer review of "Milk Consumption for the Prevention of Fragility Fractures"

_nutrients, 2020, doi:10.3390/nu12092720_

Round 1
Reviewer 1 Report
With great interest I've read the manuscript by Byberg et al on milk consumption for the prevention of fragility fractures.
In general, the paper is well written and I do not have any major comments, although some points might be clarified to improve the quality of this manuscript.
- Paragraph 2 can be shortened drastically. It basically explains why certain types of diary products cannot be compared. 75 lines is a lot.
- Clear graphs are presented (i.e. figure 1) on the intake of diary products. I am missing a clear overview regarding possible age-related differences in diary intake and subsequent influence on fragility fractures.
- Abundant background information regarding the diary products and intake/consumption is presented. However, I miss a bit more background regarding fragility fractures.
- In the table starting on line 272 on page 7, 10 articles are presented which described cohort studies of milk consumption in relation with hip fractures. Are all these studies done solely on fragility fractures of the hip, or also on 'non-fragility' fractures? And if so, what is the difference in definition for this.
Minor comment:
-Either paragraph 10 is missing, or the numbering is incorrect.
Author Response
Dear Editor and Reviewer,
Thank you for the opportunity to revise our manuscript. We have responded to the Reviewer's comments below each point, our answer is indicated by "AU", and revised the manuscript accordingly. We hope that our changes are satisfactory.
Best regards,
Liisa Byberg
Reviwer 1 comments
With great interest I've read the manuscript by Byberg et al on milk consumption for the prevention of fragility fractures.
In general, the paper is well written and I do not have any major comments, although some points might be clarified to improve the quality of this manuscript.
1. Paragraph 2 can be shortened drastically. It basically explains why certain types of diary products cannot be compared. 75 lines is a lot.
AU: We agree that this section was too long and have tried to condense the paragraph. Our experience is unfortunately that is not always obvious to readers and researchers why these products cannot be compared and we wanted to be specific with regards to the differences. We think that this may aid in future research (how to assess milk and dairy intake when planning studies and also how to handle data when analyzing data in already available studies) and for the reader when interpreting results from available studies. We hope that our changes are satisfactory.
2. Clear graphs are presented (i.e. figure 1) on the intake of diary products. I am missing a clear overview regarding possible age-related differences in diary intake and subsequent influence on fragility fractures.
AU: We have added information on age-related differences in dairy intake (paragraph 2). We chose to present the subsequent influence on fragility fractures in relation to the potential mechanisms in paragraph 4, where we also previously indicated that milk intake in childhood is related to adult height.
3. Abundant background information regarding the diary products and intake/consumption is presented. However, I miss a bit more background regarding fragility fractures.
AU: We agree, and in accordance with both Reviewers’ comments, we have now added a paragraph on fragility fractures (paragraph 3) and shortened the section on dairy products (paragraph 2).
4. In the table starting on line 272 on page 7, 10 articles are presented which described cohort studies of milk consumption in relation with hip fractures. Are all these studies done solely on fragility fractures of the hip, or also on 'non-fragility' fractures? And if so, what is the difference in definition for this.
AU: In the new paragraph 3, we describe the features of fragility fractures. We also describe that separating low energy trauma from low energy trauma fractures might be difficult and perhaps not advisable. In the table, we have now indicated the articles that report exclusion of high-energy trauma fractures from their definition of hip fracture.
Minor comment:
-Either paragraph 10 is missing, or the numbering is incorrect.
AU: Thank you for noticing this. We have updated the numbering of the paragraphs.
Reviewer 2 Report
This work has collectively presented interesting previous findings on relation between the milk consumption and risk of fragility fractures. The manuscript is well written, has good information, and should be of great interest to the readers. While, there some points that the authors should consider for the improvement of manuscript.
- Introduction part is more focused on milk part. Even though it is mentioned briefly, it would be better to include details on fragility fracture. For general readers it would be easy to understand what is meant by fragility fracture, how it occurs and various risk factors.
- Like "Definition of diary and milk intake", it would be good to have the section for "Fragility fracture". I have mentioned because, the authors have not described clearly about the term in introduction part.
- The "Potential mechanisms" section is broad but unfortunately most of the topics are not linked with fracture. The authors should focus on linking their statements mentioned in between lines 165-204 with the fracture.
- In line 343, the authors have mentioned that "Women are more likely to have suffer a hip fracture than men". It would be better if authors could add a short statement about the reason behind the women being more susceptible for fracture.
- The authors should have written what FFQ stands for.
- The authors have included very limited literature on genetic polymorphism, milk intake and fracture. Most of the previous findings have indicated no significant relationship. However, it would be better to provide the information obtained from those findings also. One of them could be PMID: 16015262.
Author Response
Dear Editor and Reviewer,
Thank you for the opportunity to revise our manuscript. We have responded to the reviewer's comments point by point below, our answers are indicated with 'AU', and revised our manuscript accordingly. We hope that our changes are satisfactory.
Best regards,
Liisa Byberg
This work has collectively presented interesting previous findings on relation between the milk consumption and risk of fragility fractures. The manuscript is well written, has good information, and should be of great interest to the readers. While, there some points that the authors should consider for the improvement of manuscript.
1. Introduction part is more focused on milk part. Even though it is mentioned briefly, it would be better to include details on fragility fracture. For general readers it would be easy to understand what is meant by fragility fracture, how it occurs and various risk factors.
AU: We agree and have added a section on Fragility fracture (paragraph 3), as also suggested below.
2. Like "Definition of diary and milk intake", it would be good to have the section for "Fragility fracture". I have mentioned because, the authors have not described clearly about the term in introduction part.
AU: A section on Fragility fracture has been added to describe this term.
3. The "Potential mechanisms" section is broad but unfortunately most of the topics are not linked with fracture. The authors should focus on linking their statements mentioned in between lines 165-204 with the fracture.
AU: Thank you for pointing this out. Our aim with this section was to give a collected overview of the different ways that milk and fermented milk products may have a different effect on inflammation and oxidative stress. We have in the paragraph before the lines you refer to tried to clarify this and the link to fracture. We also believe that the added section on fragility fracture helps to understand the link of the potential mechanisms with fragility fracture risk.
4. In line 343, the authors have mentioned that "Women are more likely to have suffer a hip fracture than men". It would be better if authors could add a short statement about the reason behind the women being more susceptible for fracture.
AU: Thank you, this information is now included in the new paragraph on Fragility fractures.
5. The authors should have written what FFQ stands for.
AU: We now explain that FFQ stands for food frequency questionnaire. Thank you for pointing this out.
6. The authors have included very limited literature on genetic polymorphism, milk intake and fracture. Most of the previous findings have indicated no significant relationship. However, it would be better to provide the information obtained from those findings also. One of them could be PMID: 16015262.
AU: Thank you for this comment and the suggested paper. The paper by Bergholdt cited in our manuscript is the only one, to the best of our knowledge, that examines the role of the lactase persistence genotype in relation to incident hip fracture, and that includes a large population. We acknowledge the suggested paper in the revised manuscript since it provides important information that self-reported lactose intolerance has a poor agreement with genetically determined lactose intolerance. The literature on lactase persistence genotype in relation to bone mineral density seems mostly based on smaller studies and are also included in the meta-analysis by Bergholdt. However, we would like to keep the focus on hip fractures as outcomes, in parallel with the discussion throughout the manuscript, and hope the amended text is satisfactory. We have also corrected the number of observations and hip fractures in the Danish study.
Round 2
Reviewer 2 Report
The authors' effort for revising the manuscript is much appreciated. Almost all the comments are addressed appropriately.
I guess there was mistyping in the line 255, that should be 5 instead of 4.